# Genetic correlations of psychiatric traits with body composition and glycemic traits are sex- and age-dependent

Christopher Hübel [1,2,3]*, Héléna A. Gaspar [1,2], Jonathan R.I. Coleman [1,2], Ken B. Hanscombe[4], Kirstin Purves [1], Inga Prokopenko [5], Mariaelisa Graff[6], Julius S. Ngwa[7,8], Tsegaselassie Workalemahu[9], ADHD Working Group of the Psychiatric Genomics Consortium, Meta-Analyses of Glucose and Insulin-related traits consortium (MAGIC), Autism Working Group of the Psychiatric Genomics Consortium, Bipolar Disorder Working Group of the Psychiatric Genomics Consortium, Eating Disorders Working Group of the Psychiatric Genomics Consortium, Major Depressive Disorder Working Group of the Psychiatric Genomics Consortium, OCD & Tourette Syndrome Working Group of the Psychiatric Genomics Consortium, PTSD Working Group of the Psychiatric Genomics Consortium, Schizophrenia Working Group of the Psychiatric Genomics Consortium, Sex Differences Cross Disorder Working Group of the Psychiatric Genomics Consortium, Substance Use Disorders Working Group of the Psychiatric Genomics Consortium, German Borderline Genomics Consortium, International Headache Genetics Consortium, Paul F. O'Reilly [1], Cynthia M. Bulik [3,10,11] & Gerome Breen [1,2]

Body composition is often altered in psychiatric disorders. Using genome-wide common genetic variation data, we calculate sex-specific genetic correlations amongst body fat %, fat mass, fat-free mass, physical activity, glycemic traits and 17 psychiatric traits (up to $N = 217,568$). Two patterns emerge: (1) anorexia nervosa, schizophrenia, obsessive-compulsive disorder, and education years are negatively genetically correlated with body fat % and fat-free mass, whereas (2) attention-deficit/hyperactivity disorder (ADHD), alcohol dependence, insomnia, and heavy smoking are positively correlated. Anorexia nervosa shows a stronger genetic correlation with body fat % in females, whereas education years is more strongly correlated with fat mass in males. Education years and ADHD show genetic overlap with childhood obesity. Mendelian randomization identifies schizophrenia, anorexia nervosa, and higher education as causal for decreased fat mass, with higher body fat % possibly being a causal risk factor for ADHD and heavy smoking. These results suggest new possibilities for targeted preventive strategies.

[1] Social, Genetic and Developmental Psychiatry Centre, Institute of Psychiatry, Psychology and Neuroscience, King's College London, London SE5 8AF, UK. [2] UK National Institute for Health Research (NIHR) Biomedical Research Centre (BRC), South London and Maudsley NHS Foundation Trust, London SE5 8AF, UK. [3] Department of Medical Epidemiology and Biostatistics, Karolinska Institutet, 171 65 Solna, Sweden. [4] Department of Medical and Molecular Genetics, King's College London, Guy's Hospital, London SE1 9RT, UK. [5] Section of Statistical Multi-Omics, Department of Clinical and Experimental Medicine, School of Biosciences and Medicine, University of Surrey, Guildford, UK. [6] Department of Epidemiology, University of North Carolina, Chapel Hill, NC 27516, USA. [7] Department of Biostatistics, Johns Hopkins Bloomberg School of Public Health, Baltimore, MD 21205, USA. [8] Department of Biostatistics, Boston University School of Public Health, Boston, MA 02118, USA. [9] Epidemiology Branch, Division of Intramural Population Health Research, Eunice Kennedy Shriver National Institute of Child Health and Human Development, National Institutes of Health, Bethesda, MD 20892, USA. [10] Department of Psychiatry, University of North Carolina at Chapel Hill, Chapel Hill 27514 NC, USA. [11] Department of Nutrition, University of North Carolina at Chapel Hill, Chapel Hill 27599 NC, USA. A full list of consortia members and their affiliations can be found in the Supplementary Information. *email: christopher.huebel@ki.se

Psychiatric disorders are complex traits influenced by thousands of genetic variants that act in concert with environmental factors[1,2]. Genome-wide association studies (GWASs) of psychiatric disorders have identified more than 300 independent genomic loci[3,4], informed biological follow-up studies[5] and may deliver promising targets for drug discovery and repurposing[6–8]. Genome-wide summary statistics generated by GWASs can be used in several different ways[9], including estimating single-nucleotide polymorphism-based heritability ($h^2_{SNP}$), which is the phenotypic variance explained by common genomic variants. Values of $h^2_{SNP}$ range from 10 to 30% for psychiatric disorders and typically capture around a third of the heritability estimated by twin studies[10]. Additionally, genetic correlations can be calculated using GWAS summary statistics via bivariate linkage disequilibrium score regression (LDSC), which estimates the genetic overlap (i.e. the shared genetic effects) between two traits[11,12]. Such GWAS-based genetic correlation analyses have shown substantial genetic overlap among psychiatric disorders[13], providing evidence for an underlying "p factor" representing general liability for psychiatric illness[14,15]. For instance, genomic structural equation modelling[16] of GWAS summary statistics for schizophrenia, bipolar disorder, major depressive disorder, post-traumatic stress disorder and anxiety showed that they load onto one shared latent factor with loading estimates between 0.29 and 0.86[16]. However, marked differences in the clinical presentation of psychiatric disorders exist for psychotic experiences or dysfunctional reward systems, suggesting the existence of additional disorder-specific genetic effects[13,14,16].

Clinically, many psychiatric disorders are accompanied by disturbances in appetite regulation, eating behaviour and altered physical activity. These disturbances can alter body composition and result in comorbid overweight or underweight[17], most prominently observed in eating disorders, such as binge-eating disorder and anorexia nervosa[18]. Such severe weight dysregulation typically reduces patients' quality of life and is associated with excess morbidity and mortality[19]. Body composition traits, including body fat % (BF%) and fat-free mass (FFM), are also complex, with substantial twin heritabilities of ~70%[20,21]. Body mass index (BMI) is the most commonly studied body composition phenotype and its associated genetic variants have been found to be significantly overrepresented in genes and genomic regions active in brain cell types[22], suggesting it may be a partially behavioural trait. Several studies have also shown negative genetic correlations of BMI with anorexia nervosa and schizophrenia[12,23–25] and positive genetic correlations of BMI with attention-deficit/hyperactivity disorder (ADHD) and major depressive disorder[26,27]. These observations suggest that an in-depth investigation of the shared genomics between psychiatric and body composition traits is needed.

In addition, both extreme overweight and extreme underweight show a clear sex difference: females are not only disproportionately affected by anorexia nervosa (with ratios up to 15:1) but also by obesity ($\geq$30 kg/m²)[28–30]. Sex differences are not limited to body composition: major depressive disorder[31] and anxiety[32] are more common in females, whereas ADHD[33] and autism spectrum disorder[34] occur more often in males. Sex differences in body composition, psychiatry and their interplay are not fully understood. Hormones and sex chromosomes have clearly been demonstrated to play a role[35], but are insufficient to fully explain the sex differences[36].

In this study, our primary aim is to identify pairs of traits with shared genetic factors by calculating sex-specific genetic correlations. To do so, we calculate sex-specific genetic correlations for GWASs of 12 psychiatric disorders mostly supplied by the Psychiatric Genomics Consortium (URLs) and five behavioural traits with sex-specific GWASs of body composition traits derived from

a healthy and medication-free subsample of the UK Biobank (URLs; Supplementary Tables 1, 2). These include BMI, BF%, absolute fat mass (FM) and FFM, as well as body composition-related traits, such as objectively measured physical activity from the UK Biobank (URLs) and glycaemic traits from MAGIC (URLs; Supplementary Data 1). We apply trait-specific illness and medication filtering to obtain genomic variants that are associated with body composition traits independent of the confounding effects of somatic diseases, such as diabetes or endocrine illnesses and addiction-related behaviours, including smoking and alcohol consumption, as well as psychiatric disorders. Where possible, putative causality is examined using generalized summary data-based Mendelian randomization (GSMR)[37] in females and males separately. As a secondary aim, we use GWASs of BMI and FFM from different stages of life, including childhood, adolescence, young adulthood and late adulthood, to identify the developmental stages in which the sharing of body composition genomic factors with genetic liability for psychiatric disorders occurs.

Here, we show that the genomic overlap between body composition traits and psychiatric disorders is evident only in later adulthood, whereas childhood and young adulthood GWASs of BMI do not correlate significantly with psychiatric traits. Accelerometer-measured physical activity shows genetic correlations with obsessive compulsive disorder (OCD) and anorexia nervosa, but with no other psychiatric disorder. In addition, glycaemic traits show significant genetic correlations only with anorexia nervosa and years of education, which positions anorexia nervosa as unique among the psychiatric disorders we investigate. These findings encourage a deeper investigation of metabolic pathways that may be implicated in psychiatric disorders to identify potential targets for preventive strategies.

## Results

**Genetic overlap between the sexes**. Body composition and physical activity showed substantial heritability explained by common genetic variation ranging from 28–51% (standard error (s.e.) = 0.4–0.8%, LDSC; Supplementary Table 3) and sex-dependent sets of genomic variation at $p_{Bonferroni} = 0.05/28 = 0.002$. We detected a genetic correlation between males and females in BF% that was significantly different from 1 ($r_g = 0.89$, s.e. = 0.03; $p_{\neq 1} = 4.7 \times 10^{-4}$, LDSC). Sensitivity analyses using Haseman–Elston regression[38] confirmed these results (Supplementary Table 3) and suggest that specific sets of genomic variation associated with BF% may be differentially active in females and males. The genetic correlations between females and males for the remaining traits are presented in Supplementary Table 4. Detailed results for the body composition and physical activity GWASs, including significant hits and Manhattan plots, are presented on Functional Mapping and Annotation (FUMA; URLs) entry 20–22 and 38–41.

**Genetic overlap of psychiatric and body composition traits**. In the genetic correlations of the psychiatric disorders and behavioural traits with body composition and physical activity, distinct patterns emerged resulting in two groups (Table 1). In the first group, anorexia nervosa, education years, OCD and schizophrenia were significantly negatively associated with BF%, while anorexia nervosa and schizophrenia were also significantly negatively associated with FFM (see Fig. 1 and Supplementary Data 2 for full results). By contrast, in the second group, ADHD, heavy smoking, alcohol dependence and insomnia were significantly positively associated with BF%, while only ADHD and heavy smoking were also significantly positively associated with FFM (Table 1). The p value threshold for the genetic correlations with body composition traits was $p_{Bonferroni} = 0.05/190 = 2.6 \times 10^{-4}$ using matrix

**Table 1 Significant genetic correlations between psychiatric disorders, behavioural traits and body composition traits.**

| Psychiatric/behavioural trait | Body composition | $r_g$ | s.e. | $p$ |
|---|---|---|---|---|
| **Group 1** | | | | |
| Anorexia nervosa | Body fat % | −0.34 | 0.03 | $2.09 \times 10^{-27}$ |
| Anorexia nervosa | Fat-free mass | −0.14 | 0.03 | $5.79 \times 10^{-06}$ |
| Education years | Body fat % | −0.34 | 0.02 | $7.11 \times 10^{-60}$ |
| Education years | Fat-free mass | −0.03 | 0.02 | 0.14 (n.s.) |
| OCD | Body fat % | −0.31 | 0.05 | $9.82 \times 10^{-10}$ |
| OCD | Fat-free mass | −0.12 | 0.04 | 0.01 (n.s.) |
| Schizophrenia | Body fat % | −0.09 | 0.02 | $7.30 \times 10^{-06}$ |
| Schizophrenia | Fat-free mass | −0.08 | 0.02 | $2.00 \times 10^{-04}$ |
| **Group 2** | | | | |
| ADHD | Body fat % | 0.30 | 0.03 | $2.50 \times 10^{-21}$ |
| ADHD | Fat-free mass | 0.17 | 0.03 | $3.84 \times 10^{-11}$ |
| Smoking | Body fat % | 0.29 | 0.03 | $3.59 \times 10^{-23}$ |
| Smoking | Fat-free mass | 0.15 | 0.03 | $9.94 \times 10^{-08}$ |
| Alcohol dependence | Body fat % | 0.23 | 0.06 | $2.00 \times 10^{-04}$ |
| Alcohol dependence | Fat-free mass | 0.04 | 0.05 | 0.45 (n.s.) |
| Insomnia | Body fat % | 0.23 | 0.04 | $2.27 \times 10^{-08}$ |
| Insomnia | Fat-free mass | 0.06 | 0.03 | 0.11 (n.s.) |

Note: The correlations were calculated using LDSC[11]. The table presents the significant findings. The full results can be found in Supplementary Data 2. Bonferroni-corrected $p$ value threshold: $a = 0.05/190 = 0.0002$

*LDSC* linkage disequilibrium score regression, $r_g$ genetic correlation, s.e. standard error, $p$ $p$ value, *n.s.* not significant after correction for multiple testing, *ADHD* attention-deficit/hyperactivity disorder, *OCD* obsessive compulsive disorder

decomposition of the genetic correlation matrix to identify the number of independent tests to adjust the threshold using Bonferroni correction[39].

**Sex differences in genetic correlations**. The genetic correlation of anorexia nervosa with BF% in females ($r_g = -0.44$, s.e. = 0.04, LDSC) was stronger than with BF% in males ($r_g = -0.26$, s.e. = 0.04, LDSC) with a significant difference of $\delta r_g = -0.17$ ($p = 4.2 \times 10^{-5}$, LDSC jackknife). Conversely, education years showed a stronger genetic correlation with FM in males than in females ($\delta r_g = 0.10$, $p = 1.3 \times 10^{-4}$, LDSC jackknife), which was also seen with FFM ($\delta r_g = 0.09$, $p = 1.7 \times 10^{-4}$, LDSC jackknife). No other sex differences were observed (Supplementary Data 3).

**Putative causal relationships**. GSMR revealed evidence consistent with putative causal relationships between psychiatric traits and body composition. The effects on continuous traits are expressed as $\beta$ coefficients (Fig. 2a–c, e, Supplementary Fig. 1), whereas the effects on binary traits are presented as odds ratios (ORs; Fig. 2d). Estimates with binary exposures were converted to the liability scale[40]. The Bonferroni-corrected $p$ value was 0.05/190 = $2.6 \times 10^{-4}$ for the GSMR analyses (Supplementary Data 4, 5). In the first group, GSMR showed evidence for a 1.8 kg decrease in FM per standard deviation of liability to anorexia nervosa ($p = 2.3 \times 10^{-8}$, GSMR) that was more pronounced in females ($\beta_{AN \to FM} = -2.14$, $p = 1.9 \times 10^{-5}$, GSMR) than in males ($\beta_{AN \to FM} = -1.3$, $p = 4.9 \times 10^{-4}$, GSMR). This mirrored the observed genetic correlations. Additionally, GSMR showed evidence for a 3.7 kg decrease in FM per year of education ($p = 5.1 \times 10^{-38}$, GSMR). Furthermore, GSMR showed a 0.88 kg decrease in FM ($p = 3.3 \times 10^{-13}$, GSMR) and a 0.58 kg decrease in FFM ($p = 4.5 \times 10^{-13}$, GSMR) per standard deviation of liability to schizophrenia (Supplementary Data 4). GSMR results for the second group showed no evidence for an influence of ADHD on fat mass ($p = 0.23$, GSMR). However, GSMR showed evidence in the reverse direction with a 1.05-fold increase in risk for ADHD per kg FM ($p = 1.3 \times 10^{-12}$, GSMR) as well as a 1.03-fold increase in risk for ADHD per kg FFM ($p = 2.0 \times 10^{-5}$, GSMR) and a

1.04-fold increase in heavy smoking per kg FM ($p = 6.7 \times 10^{-8}$, GSMR; Supplementary Data 5).

**Genetic correlations with physical activity**. In the first group, OCD ($r_g = 0.28$, s.e. = 0.07, LDSC) and anorexia nervosa ($r_g = 0.17$, s.e. = 0.05, LDSC) correlated positively with objectively measured physical activity, whereas education years showed a significant correlation with physical activity only in females ($r_g = 0.17$, s.e. = 0.04, LDSC; Supplementary Data 2). However, when formally tested the genetic correlation was not significantly different from the correlation observed in males (Supplementary Data 3). Neither ADHD ($p = 0.20$, LDSC) nor any other trait in the second group correlated with physical activity (Supplementary Data 2).

**Genetic correlations with glycaemic traits**. Our investigation into whether the relationships of the psychiatric traits with body composition are mirrored in their relationships with glycaemic traits (Fig. 3) showed that anorexia nervosa ($r_g = -0.28$; $p = 1.8 \times 10^{-7}$, LDSC) and education years ($r_g = -0.28$, $p = 1.0 \times 10^{-12}$, LDSC) correlated genetically negatively with fasting insulin concentrations. Accordingly, anorexia nervosa ($r_g = -0.29$, $p = 2.8 \times 10^{-5}$, LDSC) and education years ($r_g = -0.33$, $p = 9.2 \times 10^{-6}$, LDSC) also showed negative genetic correlations with insulin resistance. In addition, education years showed a negative genetic correlation with fasting glucose concentrations ($r_g = -0.14$; $p = 2.1 \times 10^{-5}$, LDSC), whereas heavy smoking showed a positive genetic correlation with fasting glucose concentrations ($r_g = 0.22$; $p = 2.0 \times 10^{-4}$, LDSC; Supplementary Data 6). No other psychiatric traits showed a genetic correlation with glycaemic traits passing our significance threshold.

Sensitivity analyses with female-only and male-only GWASs of the psychiatric and behavioural traits resulted in similar results, indicating that the patterns and results are consistent and largely independent of female-to-male ratios in the sex-combined GWAS (Supplementary Data 2, 6 and Supplementary Figs. 2a–3b). Sensitivity analyses not adjusting the body composition GWASs for alcohol consumption or smoking yielded the same results (Supplementary Data 7).

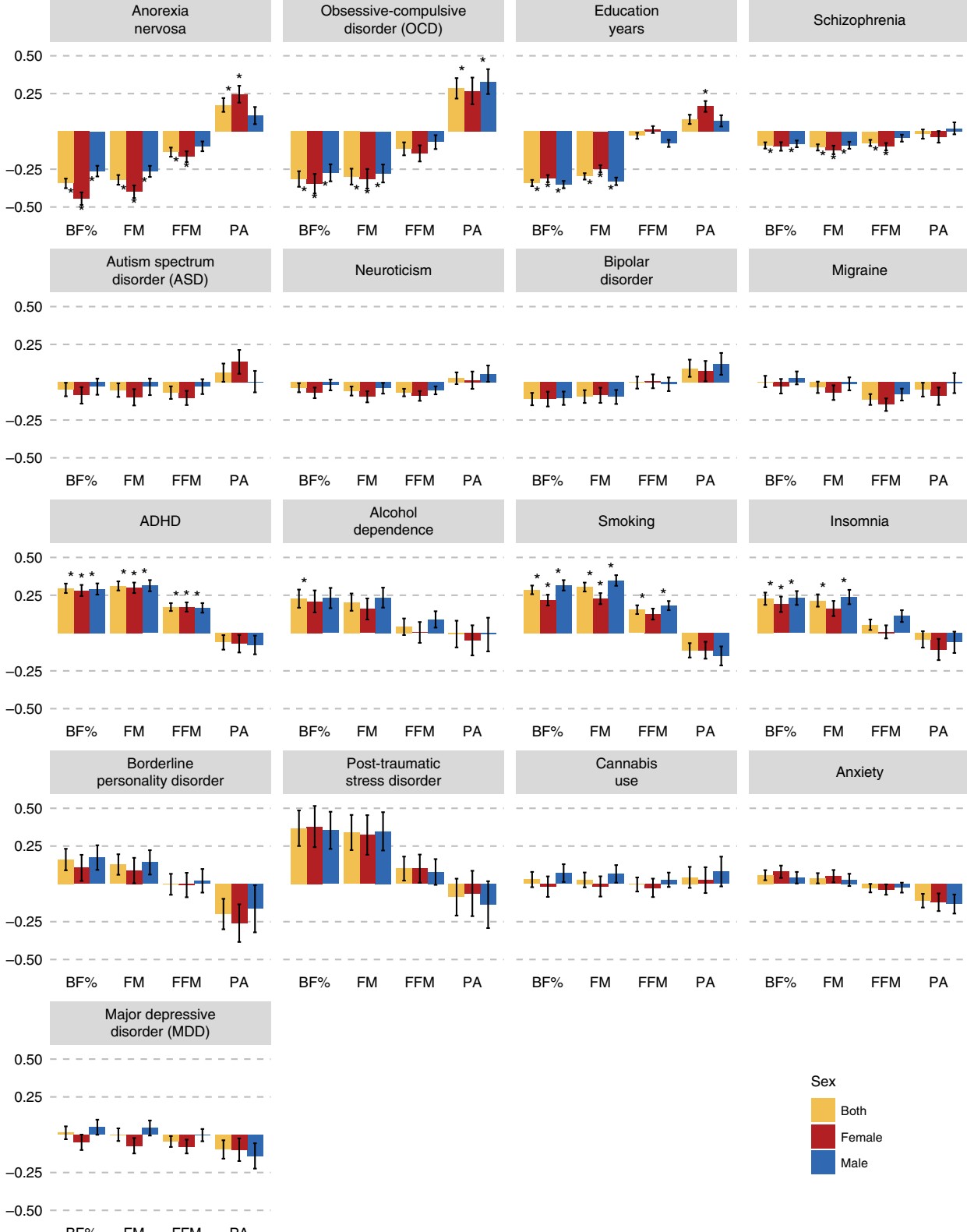

**Fig. 1 Sex-specific genetic correlations across body composition, physical activity and psychiatric traits.** Sex-specific genetic correlations of body composition traits ($n$ = up to 155,961) and physical activity ($n$ = up to 66,224) with sex-combined psychiatric disorders ($n$ = up to 77,096) and behavioural traits ($n$ = up to 157,355). The autosomal genetic correlations were calculated by bivariate linkage disequilibrium score regression (LDSC). Coloured bars represent genetic correlations, error bars depict standard errors (s.e.) and asterisks indicate statistically significant genetic correlations with $p$ values less than $\alpha = 0.0003$. This threshold was calculated via the identification of the number of independent tests using matrix decomposition of the genetic correlation matrix and subsequent Bonferroni correction of $\alpha = 0.05$ for 190 independent tests. ADHD = attention-deficit/hyperactivity disorder, BF% = body fat percentage, FFM = fat-free mass, FM = fat mass, PA = physical activity.

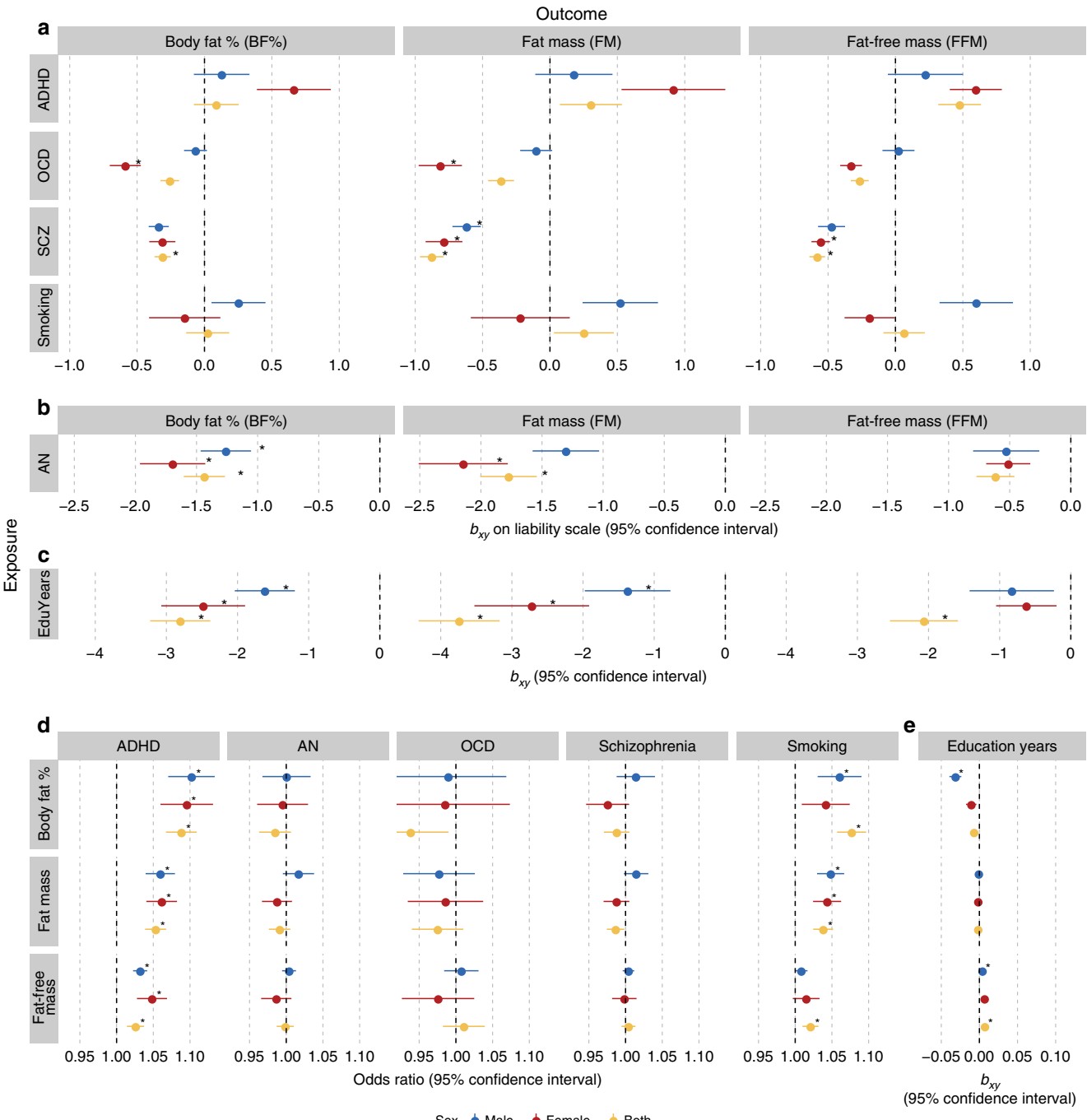

**Fig. 2 Causal associations between body composition and psychiatric traits.** Results are shown from generalized summary data-based Mendelian randomization (GSMR) analyses. Colours represent the sex of the body composition trait: red for female effects, blue for male effects and yellow for sex-combined effects. Error bars represent 95% confidence intervals (95% CIs) and asterisks indicate statistically significant estimates with *p* values less than $\alpha = 2.6 \times 10^{-4}$. **a** Putative causal associations of exposures (rows) psychiatric disorders (*n* = up to 77,096) and behavioural traits (*n* = up to 217,568) with outcomes (columns) body composition traits (*n* = up to 155,961). Dots represent the effect sizes (as measured by $\beta$, $b_{xy}$) on the liability scale of the disorders or traits. **b, c** Mendelian randomization results for the exposures anorexia nervosa and education years on the outcomes the body composition traits. These are plotted differently due to the size of the effects. All estimates are presented together in Supplementary Fig. 1 on the same scale. **d** Putative causal associations of exposures (rows) body composition traits (*n* = up to 155,961) with outcomes (columns) psychiatric disorders (*n* = up to 77,096) and behavioural traits (*n* = up to 217,568). Dots represent the effect sizes (as measured by odds ratios, ORs) of risk factors on disorders or traits. **e** The Mendelian randomization results for body composition traits as exposures on the outcome years of education. Dots represent the effect sizes (as measured by $\beta$, $b_{xy}$) on the scale of the risk factors. Abbreviations: ADHD = attention-deficit/hyperactivity disorder, AN = anorexia nervosa, BF% = body fat percentage, EduYears = education years, FFM = fat-free mass, FM = fat mass, OCD = obsessive compulsive disorder, SCZ = schizophrenia.

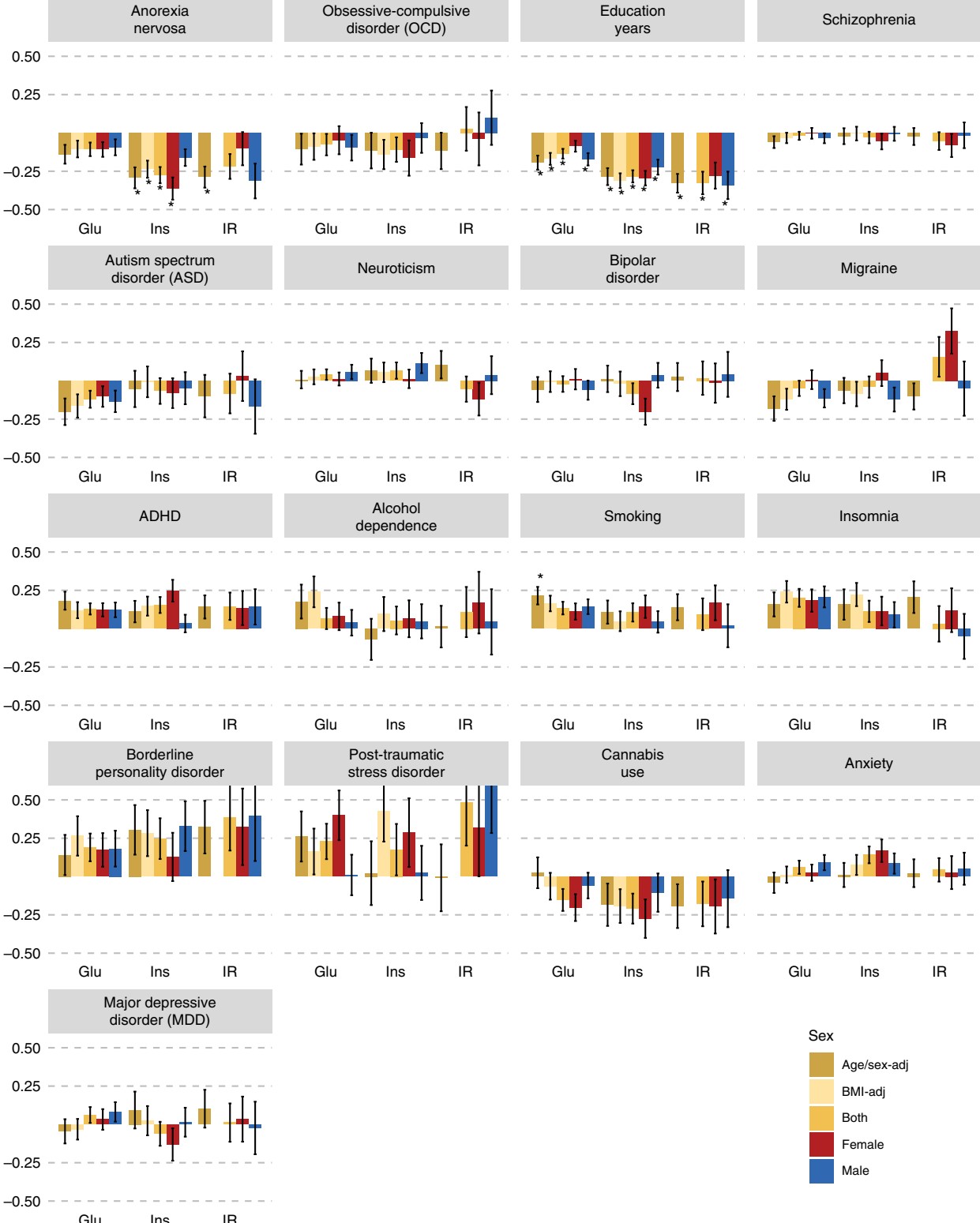

**Fig. 3 Sex-specific genetic correlations across glycaemic traits and psychiatric traits.** Sex-specific genetic correlations of glycaemic traits ($n$ = up to 140,583) with sex-combined psychiatric disorders ($n$ = up to 77,096) and behavioural traits ($n$ = up to 157,355). The autosomal genetic correlations were calculated by bivariate linkage disequilibrium score regression (LDSC). Coloured bars represent genetic correlations, error bars depict standard errors (s.e.) and asterisks indicate statistically significant genetic correlations with $p$ values less than $\alpha$ = 0.0002. This threshold was calculated via the identification of the number of independent tests using matrix decomposition of the genetic correlation matrix and subsequent Bonferroni correction of $\alpha$ = 0.05 for 231 independent tests. ADHD = attention-deficit/hyperactivity disorder, Glu = fasting glucose, Ins = fasting insulin, IR = insulin resistance, adj = adjusted.

**Age-dependent genetic correlations**. As a secondary aim, we explored the developmental dependence of genetic correlations of BMI and FFM at different ages with psychiatric disorders and behavioural traits (Fig. 4). We used BMI as a proxy measure of BF% as no GWAS of BF% in childhood or adolescence were available. To test if the sets of genetic variants affecting body composition at different stages of life differentially correlate with psychiatric disorders and behavioural traits, we estimated the following genetic BMI correlations and tested if they were significantly different from one[41]: between childhood and adolescence/young adulthood ($r_g = 1.00$, s.e. = 0.07, LDSC), between childhood and later adulthood ($r_g = 0.66$, s.e. = 0.04, LDSC) and between adolescence and later adulthood ($r_g = 0.80$, s.e. = 0.05, LDSC). The genetic correlation of FFM between childhood and adulthood was also significantly different from one ($r_g = 0.30$, s.e. = 0.04, LDSC). As above, multiple psychiatric disorders and traits showed significant positive and negative genetic correlations with adult BMI and FFM. However, neither BMI in childhood, adolescence or young adulthood, nor FFM in childhood, showed significant genetic correlations with any of the psychiatric disorders or behavioural traits (Supplementary Data 8). To additionally test an extreme phenotype, we calculated genetic correlations between psychiatric traits and obesity in childhood. Within the first group, only education years ($r_g = -0.19$, s.e. = 0.03, LDSC) correlated negatively with obesity in childhood. In the second group, ADHD was the only psychiatric disorder that showed a significant positive genetic correlation with obesity in childhood ($r_g = 0.22$, s.e. = 0.05, LDSC). GSMR analyses gave evidence for a 1.42-fold increase for ADHD per kg/m² increase in childhood BMI ($p = 1.26 \times 10^{-8}$, GSMR).

## Discussion

Symptomatically, psychiatric disorders are often accompanied by alterations in energy intake, energy expenditure and body composition. Recent genetic analyses of BMI found an important role for genes expressed in the brain and specific brain cell types[22], suggesting that BMI may be a metabo-behavioural trait. This spurred our in-depth investigation of the shared genetics of psychiatric traits and body composition. We were able to show that five psychiatric disorders—anorexia nervosa, OCD, schizophrenia, ADHD and alcohol dependence—as well as three behavioural traits—education years, insomnia and heavy smoking—show significant genetic correlations (i.e. shared genetics) with body composition in two distinct patterns.

The first group of psychiatric disorders and behavioural traits included anorexia nervosa, OCD, schizophrenia and education years, and was characterized by genetic correlations with genomic variants predisposing to lower BF% and FFM. The second group comprised ADHD, alcohol dependence, heavy smoking and insomnia, and had genetic correlations with genomic variants predisposing to higher BF% and FFM. Our Mendelian randomization analyses used significant genetic variants as instrumental variables and found that anorexia nervosa, schizophrenia and education years showed evidence consistent with a negative causal effect on FM and, in the reverse direction, higher BF% appeared to be a risk factor for both ADHD and heavy smoking. Our results also suggested that the overweight seen in individuals with schizophrenia in epidemiological studies[42] is likely to represent medication effects[43] given our observations of a putative causal effect of schizophrenia on lower FM and FFM. This finding reiterates the pressing need for the development of new antipsychotic medications with more favourable weight-related side effect profiles.

In our analysis, anorexia nervosa showed a stronger correlation with BF% in females than in males. This phenomenon was not observed for other traits genetically associated with anorexia nervosa, such as neuroticism, anxiety, major depressive disorder, OCD or schizophrenia[41]. These findings suggest that anorexia nervosa and BF% may share a sex-dependent set of genomic variants potentially contributing to the marked sex bias in the prevalence of anorexia nervosa. Education years showed a stronger genetic correlation with FM in males than in females. However, the GSMR analysis showed a more pronounced protective effect of education years on FM in females than in males in line with a large epidemiological study[44]. This suggests that the stronger genetic association between education years and FM in males may be driven by a set of pleiotropic variants.

From a developmental perspective, it is striking that GWASs of body composition across ages do not genetically correlate perfectly with each other. These varying genetic effects across the lifespan[41,45] have been termed "genetic innovation"[46] and represent the effects of partially different, age-dependent sets of genomic variants on body composition regulation at certain periods of life[41,45]. Some of the psychiatric disorders, such as ADHD and anorexia nervosa, typically have their onset in childhood or adolescence with preceding symptoms or behaviours that implicate neurodevelopmental components. We used the available life-stage GWASs of body composition and did not find genetic overlap between childhood or adolescence/young adulthood BMI with psychiatric disorders, but instead found significant genetic correlations of psychiatric disorders with later adult BMI and BF%. Our analyses also show that genetic variants associated with obesity before the age of ten were positively correlated only with ADHD and negatively only with education years. The relatively specific positive genetic correlation of childhood obesity with ADHD recapitulates a large body of clinical evidence of high phenotypic comorbidity[47], also shown in family studies[48]. Overweight may represent a difficult but potentially intervenable risk factor at a young age.

Our finding of a genetic overlap between ADHD and obesity in childhood may implicate shared biological pathways between both traits. Given our other results, it appears that this shared component is unlikely to be related to physical activity or glycaemic traits. Instead, speculatively, a central nervous system pathway that is dysregulated by increased body mass in childhood may increase the liability to develop ADHD.

We also investigated body composition-related traits, including physical activity, fasting insulin and fasting glucose concentrations. Physical activity showed a positive genetic correlation with anorexia nervosa and OCD, which themselves were negatively genetically correlated with BF%. Carrying genetic variants that predispose to higher physical activity may be associated with the relationship between lower BF% and psychiatric traits. Higher physical activity, therefore, should be carefully assessed in the treatment of patients with compulsive psychiatric disorders like anorexia nervosa and OCD as it may be a genetically mediated behaviour, as indicated by our analysis.

Contrary to our expectations, ADHD did not show a genetic correlation with physical activity. This suggests that hyperactivity in ADHD may not originate from biological liability for higher accelerometer-measured physical activity[49] and is likely to have an alternative cause, such as insufficient inhibitory control as observed in paediatric clinical samples with ADHD[50], healthy adult population samples[51], and in a large longitudinal developmental cohort study[52].

Our analyses showed that anorexia nervosa and education years have a negative genetic correlation with fasting insulin concentrations and insulin resistance, positioning anorexia nervosa as a special case within the psychiatric disorders and potentially differentiating it from OCD. These negative correlations with fasting insulin concentrations mirrored the negative

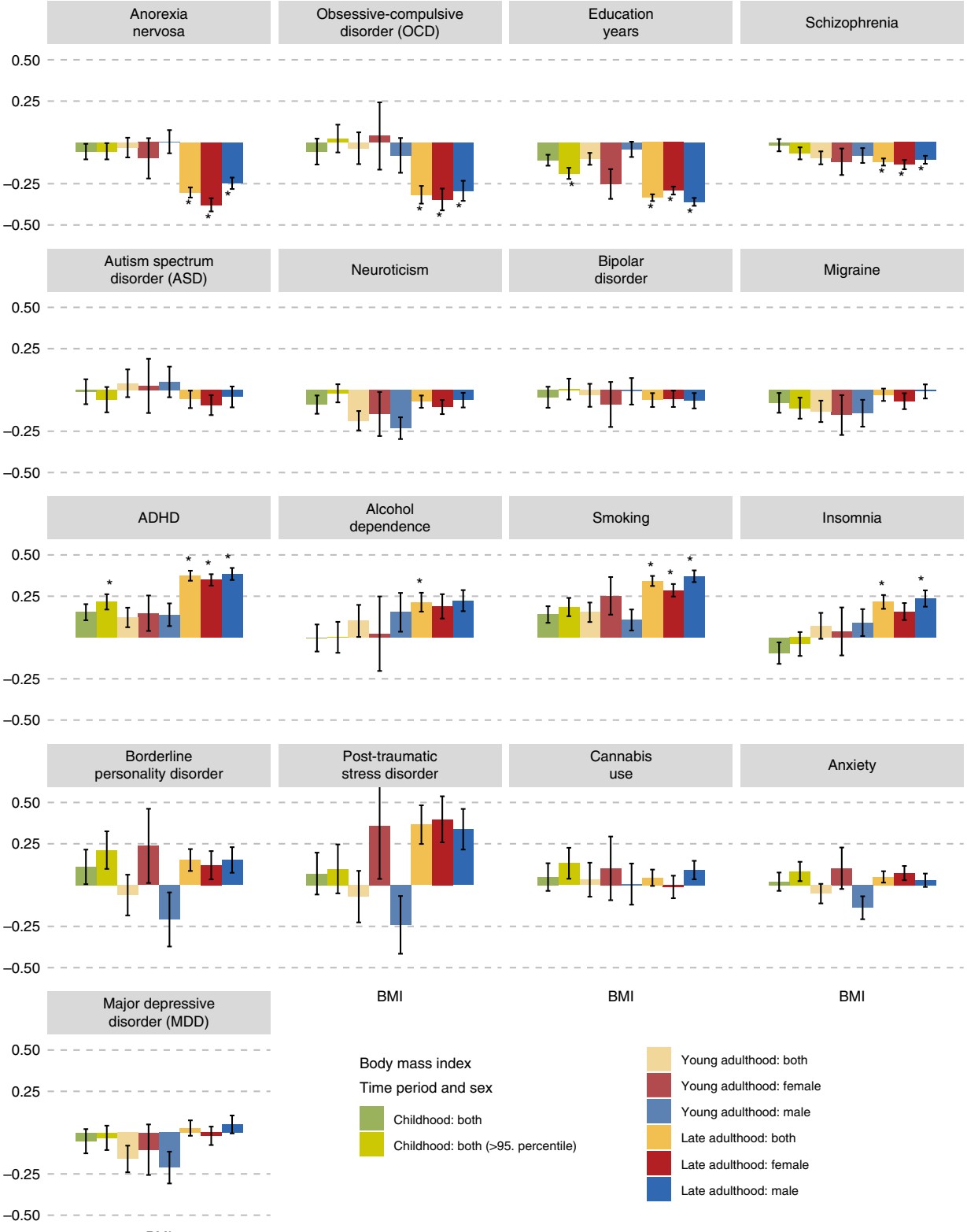

**Fig. 4 Age-dependence of sex-specific genetic correlations across body composition and psychiatric traits.** Sex-specific genetic correlations of body mass index (BMI) and fat-free mass (n = up to 157,355) with psychiatric disorders (n = up to 77,096) and behavioural traits (n = up to 157,355) across the lifespan. Participants of the childhood BMI GWAS (green, n = 35,668) were younger than 10 years, the participants of the young adulthood GWASs (lighter colours, n = 29,054) were between 15 and 35 years, participants of the late adulthood GWAS (darker colours, n = 155,961) were between 39–75 years old. Overweight in childhood (lime green; n = 13,848) was included as an extreme phenotype. The autosomal genetic correlations were calculated by bivariate linkage disequilibrium score regression (LDSC). Coloured bars represent genetic correlations, error bars depict standard errors (s.e.) and asterisks indicate statistically significant genetic correlations with p values less than α = 0.0002. This threshold was calculated via the identification of the number of independent tests using matrix decomposition of the genetic correlation matrix and subsequent Bonferroni correction of α = 0.05 for 210 independent tests. ADHD = attention-deficit/hyperactivity disorder.

genetic correlations between anorexia nervosa, education years and BF%. The potential involvement of metabolic hormones like insulin in anorexia nervosa underscores the relationship of brain and body and their reciprocal regulation[53], opening an avenue for deeper investigation of metabolic components in psychiatric disorders. The genetic correlations of ADHD with glycaemic traits were not significant, implying that these traits play a smaller role in ADHD than in anorexia nervosa, given the comparable sample size of the GWASs on both psychiatric disorders[25,26]. Genetic associations of physical activity and glycaemic traits with body composition and psychiatric traits in plausible directions render them interesting candidates for formal mediation analyses as they may be actionable targets[54].

Our study represents the largest investigation of sex- and age-dependent effects in the genomic overlap of body composition and psychiatric traits. Although our analyses drew on the largest available GWASs, some phenotypes still had relatively small sample sizes for genomic investigations of common variants in complex traits, especially for our sex-specific analyses. These should be repeated when sample sizes have increased, especially for OCD as its currently available GWAS sample size is particularly modest. All Mendelian randomization analyses, using GSMR[37], with body composition or glycaemic traits, ADHD, education years, schizophrenia or heavy smoking as exposure were sufficiently powered; however, the analyses with anorexia nervosa, insomnia or OCD as exposures should be regarded as exploratory in nature because p value thresholds were lowered to include at least 10 single-nucleotide polymorphisms (SNPs) in the instrument variable.

Finally, the age-dependent genetic influences we observed between psychiatric traits and body composition suggests that future research could focus on a developmental approach to GWAS analyses of body composition, to capture age- and sex-dependent differences. These differences have already been suggested by larger twin studies[55,56] and two molecular genetic studies[41,45], which enabled our examination of their relationship with psychiatric traits. Most importantly, shared biological pathways and common environmental factors influencing both body composition and behavioural traits should be studied as potential targets for interventions.

## Methods

**UK Biobank subsample**. We performed GWASs on an unrelated (KING relatedness metric >0.044, equivalent to a relatedness value of 0.088; $n_{related} = 7765$) European subsample (defined by 4-means clustering of the genetic principal components)[57] of the genotyped UK Biobank participants ($n = 155,961$, 45% female, 32% of the genotyped participants, Supplementary Table 1)[58,59]. The UK Biobank (URLs) is a prospective cohort sampled from the general population between 2006 and 2010. All participants were between 40 and 69 years old, were registered with a general practitioner through the United Kingdom's National Health Service, and lived within travelling distance of one of the assessment centres.

**Ethics**. The UK Biobank is approved by the North West Multi-centre Research Ethics Committee. All procedures performed in studies involving human participants were in accordance with the ethical standards of the North West Multi-centre Research Ethics Committee and with the 1964 Declaration of Helsinki and its later amendments or comparable ethical standards. All participants provided written informed consent to participate in the study. This study has been completed under UK Biobank approved study application 27546.

**Power calculations of the GWASs**. We conducted power calculations for the female and male GWASs using the Genetic Power Calculator[60]. A minimum of 39,580 individuals is required to detect a SNP that accounts for 0.1% of trait variance at 80% power at a genome-wide significance threshold of $p \leq 5 \times 10^{-8}$ and a minor allele frequency of 0.20. According to these results, the female and the male GWASs were sufficiently powered to detect genome-wide significant loci with 70,700 females and 85,261 males. With these parameters, the female GWAS had a power of 99.8% and the male GWAS of 99.9%.

**GWASs on body composition traits in the UK Biobank**. The continuous body composition traits—BF%, FM, FFM and BMI—were measured using the validated bioelectrical impedance analyser Tanita BC-418 MA (Tanita Corporation, Arlington Height, IL) at every assessment centre[61,62] for every participant across the UK. We applied trait-specific medication and illness filtering to exclude participants with compromised hydration status and medications or illnesses known to affect body composition to identify genetic variation associated with body composition phenotypes that is not confounded by illnesses and their downstream effects or metabolism-changing medication. We applied stringent exclusion criteria and covaried for addictive behaviour-related phenotypes, including smoking and alcohol consumption (for exclusion criteria, see Supplementary Table 2). We regressed the body composition traits on factors related to assessment centre, genotyping batch, smoking status, alcohol consumption, menopause and continuous measures of age, and socioeconomic status (SES) measured by the Townsend Deprivation Index[63] as independent variables. We took the residuals from these regressions as our phenotypes for the GWASs. We included 7,794,483 SNPs and insertion–deletion variants (hereafter referred to as SNPs) with a minor allele frequency >1%, imputation quality scores >0.8, and that were genotyped, or present in the HRC reference panel[64] and used an additive model on the imputed dosage data provided by UK Biobank, using BGENIE v1.2[65]. We accounted for underlying population stratification by including the first six principal components, calculated on the genotypes of our European subsample using FlashPCA2[66]. We performed GWASs including incremental numbers of principal components and checked each GWAS for inflation by calculating its LDSC intercept. We identified six principal components as the optimal number to adjust for population stratification within the European subsample and to not overcorrect the analysis retaining the greatest signal. Additionally, we included assessment centre as a covariate to adjust for population stratification. We then meta-analysed the sex-specific GWASs using METAL[67] (URLs) applying an inverse variance-weighted model with a fixed effect, to obtain sex-combined results.

**Clumping and genome-wide significant loci**. Significantly associated SNPs ($p < 5 \times 10^{-8}$) were considered as potential index SNPs. SNPs in LD ($r^2 > 0.2$) with a more strongly associated SNP within 3000 kb were assigned to the same locus using FUMA (URLs)[68]. Overlapping clumps were merged with a second clumping procedure in FUMA, merging all lead SNPs with $r^2 = 0.1$ to genomic loci. After clumping, independent genome-wide significant loci ($5 \times 10^{-8}$) were compared with entries in the NHGRI-EBI GWAS catalogue[69], using FUMA[68].

**Heritability estimation and investigation of sex differences**. To ensure the robustness of our results, we applied multiple approaches to calculate heritability estimates and genetic correlations. We used BOLT-LMM[70], LDSC[11] and GREML[71] implemented in GCTA[72] to calculate common variant $h^2_{SNP}$ (URLs). Additionally, we calculated the genetic correlation between females and males using LDSC[11] and Haseman–Elston regression[38] implemented in GCTA[72] to estimate sex differences in the genetic architecture of the body composition, glycaemic traits and physical activity. Haseman–Elston regression uses the cross-product of phenotypes for pairwise individuals and a genetic relatedness matrix to calculate heritability and genetic correlations[73]. All other statistics were calculated in R 3.4.1 if not otherwise stated (URLs).

**GWASs of psychiatric disorders and behavioural traits**. All of the following traits were used for the sex-specific and age-dependent analyses (Supplementary Data 1). The sex-specific summary statistics for the psychiatric disorders, including major depressive disorder[27], schizophrenia[3], anorexia nervosa[25], bipolar disorder[74,75], ADHD[26,76], alcohol dependence[77], autism spectrum disorder[78] and PTSD[79], were provided by the PGC (URLs), for OCD[80,81] by International Obsessive Compulsive Disorder Foundation Genetics Collaborative (IOCDF-GC) and OCD Collaborative Genetics Association Studies (OCGAS), for borderline personality disorder[82] by the German Borderline Genomics Consortium, for cannabis use by the International Cannabis Consortium[83], for anxiety[84] by our own group, for insomnia[85] by the Complex Trait Genetics group at VU University Amsterdam (URLs), for heavy smoking[86] by University of Leicester available from the UK Biobank (URLs), for the behavioural traits years of education[87] by the Social Science Genetic Association Consortium (SSGAC) (URLs), for neuroticism[41] by our own group (Supplementary Data 1) and for migraine[88,89] by International Headache Genetics Consortium (IHGC). Glycaemic traits'[90] summary statistics were provided by the Meta-Analyses of Glucose and Insulin-related traits Consortium (MAGIC), whereas childhood obesity[91] results were provided by the Early Growth Genetics (EGG, URLs) Consortium, BMI in young adulthood by Graff et al.[92] and physical activity by our group[41].

**Genetic correlations**. Using an analytic extension of LDSC[11], we calculated SNP-based bivariate genetic correlations ($r_g$) to examine the genetic overlap of body composition and glycaemic traits with psychiatric and behavioural traits and disorders in a sex-specific manner. Differences in genetic correlations were calculated and their s.e.'s were calculated using a block jackknife approach as previously described[41].

**Generalized summary data-based Mendelian randomization**. We investigated putative causal bidirectional relationships between these traits using GSMR[37]. Mendelian randomization is a method that uses genetic variants as instrumental variables, which are expected to be independent of confounding factors, to test for causative associations between an exposure and an outcome[93]. Mendelian randomization can be used to infer credible causal associations when randomized-controlled trials are not feasible or are unethical[93]. GSMR performs a multi-SNP Mendelian randomization analysis using summary statistics. Let $z$ be a genetic variant (e.g. SNP), $x$ be the exposure (e.g. psychiatric disorder) and $y$ be the outcome (e.g. body composition trait). First, GSMR is based on the premise that several nearly independent SNPs ($z$) are associated with the exposure ($x$). Second, it assumes that the exposure ($x$) has a causal effect on $y$. If both assumptions hold true, the SNPs that are associated with the exposure ($x$) will exert an effect on the outcome ($y$) via the exposure ($x$). If in this instance no pleiotropy is present, the estimate ($b_{xy}$) at any of the SNPs that are associated with the exposure ($x$) should be highly similar, because each effect of all SNPs on the outcome ($y$) will be mediated through the exposure ($x$). With the help of a generalized least squares (GLS) model, the estimates of $b_{xy}$ of each SNP that is associated with the exposure ($x$) can be combined, resulting in higher statistical power[37,94]. The GSMR method essentially implements summary data-based Mendelian randomization analysis for each SNP instrument individually, and integrates the $b_{xy}$ estimates of all the SNP instruments by GLS, accounting for the sampling variance in both $b_{zx}$ and $b_{yz}$ for each SNP and the LD among SNPs. We used individual-level genotype data from a subsample of the anorexia nervosa GWAS to approximate the underlying LD structure to account for LD between the variants in the multi-SNP instrument. Pleiotropy is an important potential confounding factor that could bias the estimate and often results in an inflated test statistic in Mendelian randomization analysis. We also removed potentially pleiotropic SNPs (i.e. SNPs that have effects on both risk factor and outcome) from this analysis using the heterogeneity in dependent instruments outlier method[37,95] that detects pleiotropic SNPs at which the estimates of $b_{xy}$ are significantly different from expected under a causal model. The power of detecting a pleiotropic SNP depends on the sample sizes of the GWAS data sets and the deviation of $b_{xy}$ estimated at the pleiotropic SNP from the causal model. Based on this, the overall $b_{xy}$ can be estimated from all the instruments remaining using a GLS approach that takes the LD between the variants and the correlations between their effect sizes into account by modelling them in a covariance matrix. Additionally, GSMR uses the intercept of the bivariate LD score regression to account for potential sample overlap between the GWASs used as instruments for the exposure or outcome[12]. Estimates with binary exposures were converted to the liability scale[40]. Some of these analyses are exploratory because a few utilised GWASs were underpowered (i.e. did not detect ≥10 genome-wide significant independent loci at a $p$ value level of $5 \times 10^{-8}$) and we therefore lowered the $p$ value threshold for inclusion, in order to include at least 10 independent SNP instruments as previously recommended[37].

**Correction for multiple testing**. We calculated the number of independent traits by matrix decomposition (i.e. number of principal components accounting for 99.5% of variance explained) and adjusted our $p$ value threshold accordingly. The first matrix of the main analysis contained all 17 psychiatric traits, all four body composition traits, physical activity and childhood obesity (Supplementary Data 2). All sex-specific correlations were entered when available. The second matrix comprised all 17 psychiatric traits and all glycaemic traits listed in Supplementary Data 6, including their sex-specific correlations. The family-wise Bonferroni-corrected $p$ value threshold for the main analysis, including the genetic correlations with body composition traits and physical activity, was $p_{Bonferroni} = 0.05/190 = 2.6 \times 10^{-4}$ and the family-wise $p$ value threshold for the genetic correlations with glycaemic traits was $p_{Bonferroni} = 0.05/231 = 2.2 \times 10^{-4}$.

**URLs**. For METAL, see http://csg.sph.umich.edu/abecasis/metal/; for FUMA, see http://fuma.ctglab.nl/; for SSGAC, see https://www.thessgac.org/; for Complex Traits Genetics lab, see https://ctg.cncr.nl/; for International Headache Genetics Consortium, see http://www.headachegenetics.org/; for the MAGIC, see https://www.magicinvestigators.org/; for UK Biobank, see https://www.ukbiobank.ac.uk/; for the PTSD working group of the Psychiatric Genomics Consortium, see https://pgc-ptsd.com/; for the Psychiatric Genomics Consortium, see http://www.med.unc.edu/pgc; for the R project, see https://www.r-project.org/; for the EGG Consortium, see https://egg-consortium.org/.

**Reporting summary**. Further information on research design is available in the Nature Research Reporting Summary linked to this article.

## Data availability

Supplementary Data 1 contains all information on data availability, including download links for summary statistics. Summary statistics for the body composition GWASs are available at www.topherhuebel.com/GWAS and the GWAS catalogue (www.ebi.ac.uk/gwas/). All sex-combined summary statistics for the psychiatric disorders are available at www.med.unc.edu/pgc/results-and-downloads/ and for glycaemic traits at https://www.magicinvestigators.org/. Sex-specific summary statistics of the psychiatric disorders can be requested from each working group of the Psychiatric Genomics Consortium by submitting a secondary analysis proposal. The data that support the findings of this study are available from UK Biobank (www.ukbiobank.ac.uk). Restrictions apply to the availability of these data, which were used under license for the current study (Project ID: 27546). Data are available for bona fide researchers upon application to the UK Biobank.

## Code availability

Analysis code can be accessed on github.com/topherhuebel/ukbgwas. Software can be accessed for BGENIE, at https://jmarchini.org/bgenie/; for BOLT-LMM v2.3.2, at https://data.broadinstitute.org/alkesgroup/BOLT-LMM/; for LDSC, at v1 https://github.com/bulik/ldsc; for METAL, at http://csg.sph.umich.edu/abecasis/metal/; for, at R 3.4 https://www.r-project.org/; for GSMR, at https://cnsgenomics.com/software/gcta/

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

## Acknowledgements

This study represents independent research part funded by the UK National Institute for Health Research (NIHR) Biomedical Research Centre at South London and Maudsley NHS Foundation Trust and King's College London. The views expressed are those of the author(s) and not necessarily those of the UK NHS, the NIHR or the Department of Health and Social Care. High-performance computing facilities were funded with capital equipment grants from the GSTT Charity (TR130505) and Maudsley Charity (980). Research reported in this publication was supported by the USA National Institute of Mental Health of the National Institutes of Health (NIMH) under Award Number U01 MH109514, U01 MH109528, U01 MH109514 and U01 MH109536. The PGC Substance Use Disorders group acknowledges support from MH109532. The content is solely the responsibility of the authors and does not necessarily represent the official views of the National Institutes of Health. Prof. Bulik acknowledges funding from the Swedish Research Council (VR Dnr: 538-2013-8864) and the Klarman Family Foundation (the Anorexia Nervosa Genetics Initiative is an initiative of the Klarman Family Foundation). Profs. Bulik and Micali are supported by NIMH R21 MH115397. PFO receives funding from the UK Medical Research Council (MR/N015746/1) and the Wellcome Trust (109863/Z/15/Z). Dr. Graff acknowledges funding from the National Institutes of Health (R01HD057194). Dr. Workalemahu acknowledges funding by the Intramural Research Programme of the Eunice Kennedy Shriver National Institute of Child Health and Human Development, National Institutes of Health. Dr. Prokopenko was funded by the European Union's Horizon 2020 research, and innovation programme (LONGITOOLS, H2020-SC1-2019-874739; DYNAhealth, H2020-PHC-2014-633595); and the Wellcome Trust (WT205915). Data on glycaemic traits have been contributed by MAGIC investigators and have been downloaded from www.magicinvestigators.org. Data on the childhood BMI trait has been contributed by the EGG Consortium and has been downloaded from www.egg-consortium.org. This study was completed as part of approved UK Biobank study application 27546 to Dr. Breen. Open access funding provided by Karolinska Institute.

## Authors contributions

C.H., C.M.B. and G.B. designed research; C.H., H.A.G., J.R.I.C., K.B.H., K.P., I.P., M.G., J.S.N. and T.W. provided essential materials; C.H., H.A.G., J.R.I.C., K.B.H. and K.P. analysed data or performed statistical analysis; C.H., H.A.G., J.R.I.C. and G.B. wrote paper; C.H. and G.B. had primary responsibility for final content. All authors read and approved the final manuscript.

## Competing interests

Dr. Breen has received grant funding from and served as a consultant to Eli Lilly, has received honoraria from Illumina and has served on advisory boards for Otsuka. Dr. Bulik is a grant recipient from and has served on advisory boards for Shire. She receives royalties from Pearson. All interests are unrelated to this work. Dr. Coleman, Dr. Gaspar, Ms. Purves, Dr. Hübel, Dr. Hanscombe, Dr. Prokopenko, Dr. Graff, Dr. Ngwa, Dr. Workalemahu and Dr. O'Reilly declare no competing interests.
