## [Peer Review File · Nature Communications]

Reviewers' Comments:

Reviewer #1:

Remarks to the Author:

Using data from the UK biobank (n=155,961; 45% female), sex-specific genetic correlations were calculated between psychiatric disorders and related behavioral traits with metrics of body composition and related traits. Results largely replicate prior work showing that genomic risk for AN, SCZ, OCD, EDU are negatively correlated with body fat and fat free mass while genomic risk for ADHD, AD, insomnia, and heavy smoking are positively correlated with body fat, fat free mass. The study extends prior research in this area by evaluating a large number of trait correlations beyond what has been previously done and incorporating sex-specific analyses. For example, a stronger connection between AN and body fat in females while EDU showed a higher relationship in males are reported. MR analyses provide evidence for a potential causal impact of SCZ, AN, and EDU on body mass while body fat may contribute to ADHD and heavy smoking but protect against AN. The study is interesting and incrementally adds to our understanding of body composition and psychiatric disorders. My major concerns involve the analytic methods (point 1 specifically).

1. Covariates. Covariates in the GWASs of body composition and related traits included the following covariates: 1) assessment centre, genotyping batch, smoking status, alcohol consumption, menopause, age, SES, 6 ancestral PCs. The rationalization for these covariates is unclear. In particular, it is unclear why variables such as smoking status and alcohol consumption are included. This would seem to be most clearly problematic for genomic correlation analyses subsequently evaluating GWASs of very clearly related phenotypes (e.g., e.g., smoking status covariate – genomic correlation with heavy smoking, alcohol consumption covariate – genomic correlation with alcohol dependence, SES and EDU). However, one could argue they may also be problematic for other covariates – e.g., might smoking be a mechanism linking psychiatric disorder to body composition? This appears inappropriate – e.g., it does not make sense to regress out the effects of alcohol consumption on body fat when examining genomic correlations between alcohol dependence and body fat - what is the genetic correlation between AD and body fat once one removes alcohol consumption? The authors are encouraged to rerun these analyses with PCs, age, genotyping batch, menopause, and assessment centre as covariates.

2. Given that many other studies of the UK biobank use 20 PCs, justification for using 6 is needed.

3. MR Pleiotropy Additional information should be provided on how the GSMR mendelian randomization analyses were conducted and in particular how these analyses addressed pleiotropy (For example was HEIDI-outlier used?).

4. Minor comment: The adjusted Bonferroni multiple correction testing for body composition/physical activity and glycemic traits analyses independently appear appropriate given the correlated nature of many interested variables. However, additional description/clarification i.e., how many independent traits went into the matrix decomposition, would be useful.

Reviewer #2:

Remarks to the Author:

The manuscript by Hübel et al. investigated the genetic relationships between psychiatry disorders/behavioral traits and energy homeostasis/body composition traits. The authors utilized several GWAS summary statistics to explore shared genetic factors between the selected traits. The strength of the study is that the authors calculated sex-specific common variant genetic correlations of body composition with psychiatric and behavioral traits. The work has been performed quite comprehensively and these findings generate new hypotheses for targeted preventive strategies. Below are my main comments followed by some minor comments.

1. The authors only mentioned sex-specific common variant genetic correlations of body composition, physical activity and glycemic traits with 17 psychiatric and behavioral traits in the abstract section, but did not mention the age-dependent genetic influences they observed between psychiatric traits and body composition. I would suggest that they should add a brief summary description in this section.
2. The study is generally a statistics study which pinpoint the shared genetics between psychiatry disorders/behavioral traits and energy homeostasis/body composition traits. And the authors do find some associations between these traits. However, the authors should provide more biological interpretation in the discussion part. For example, ADHD have high genetic correlation with body fat %, the question is are there some shared biological pathways that responsible for this phenomenon?
3. The introduction: On line 79 and 80, the authors stated that "...These analyses have shown substantial overlap between psychiatric disorders, providing evidence for an underlying "p factor" representing general liability for psychiatric illness ...", the authors introduced several different approaches to use genome wide summary statistics generated by GWAS but it is not clear what are substantial overlaps between psychiatric disorders. It should be clarified simply or provide some examples to demonstrate it.
4. Page 422: Section of "GWAS of psychiatric disorders and behavioral traits", it should be clarified in the online methods section, which psychiatric disorders and behavioral traits GWASs were used for sex- and age-dependent genetic correlations analysis.
5. More details on the generalized summary data-based Mendelian randomization (GSMR) methods are required in the online methods (on page 446). Specifically, it would be good to add a description of the principle of GSMR and the meaning of the parameters.
6. Now it reads as slightly too comprehensive and does not give a reality check. You just were able to show that five psychiatric disorders (anorexia nervosa, OCD, schizophrenia, ADHD, and alcohol dependence) as well as three behavioral traits (education years, insomnia, and heavy smoking) have significant genetic correlations (i.e., shared genetics) with body composition in two distinct patterns, however, the main drawback with the current version of the manuscript is that it is not easy to understand what you actually found or what are the specific applications of these genetic correlations. And the Table 1 tries to summarize this, but is so comprehensive that it does not highlight the main findings. This should be clarified.

Minor comments

1. The current version of this manuscript needs to be checked for logic and you need to pay attention to the typesetting of the text. For example on page 411 the authors forgot the blank line.
2. We try to get access to the URL <https://github.com/topherhuebel>, but the code used for analysis in this research is not uploaded yet. It is highly recommended to upload the source code for further review.

Reviewer #3:

Remarks to the Author:

This article conducts a cross-trait analysis of summary statistics for psychiatric disorders, body composition, glycemic traits and physical activity, using LDSR for genetic correlation, GSMR for mendelian randomization, and several other well-established methods for the supplementary analyses. The manuscript is very well written, and the results provide novel scientific insights. Particularly, it is intriguing that the genetic correlation between schizophrenia and body fat% is negative, against what one would expect from the epidemiological studies. Similarly, it is interesting that there is a genetic relationship between ADHD/EDU and overweight before the age of ten, and a lack of genetic correlation between ADHD and physical activity. These findings are well interpreted in the discussion section and might have important clinical implications.

I only have a few comments:

1. Please revise Fig2 and the corresponding description of GSMR results, as it has several inconsistencies:

- Line 171: "1.9 kg decrease in fat mass per year of education", is this statement correct? Here 1.9 seems like a very large effect, at least if EDU is measured in the actual years (which I would expect to range somewhere from 10 to about 20)?
- Fig2C, EDU years are shown as "odds ratio" which is inconsistent with Supplementary Table 3 where EDU is shown as a continuous trait.
- Lines 163-165: The description says "Odds Ratio (Fig 2A, 2B), Beta (Fig 2C)" which is inconsistent with the actual Fig2, which shows Beta on Fig 2A, 2B, and Odds Ratio on Fig 2C

2. It would be helpful to expand the discussion around sex-specific genetic correlations, i.e. Anorexia nervosa (AN) and Years of education (EDU) versus body fat%. Is there some way to interpret these findings? If I understood correctly, the results stand for correlation of sex-specific body fat% versus non-sex-specific AN and EDU. Would it be possible to conduct a similar analysis of sex-specific EDU and sex-specific AN versus (also sex-specific) body fat% ?

3. I suggest adding few words about the limitations of the GSMR. For example, is there sample overlap between GWAS studies in the analysis that may affect GSMR results? Is there an indication of the pleiotropic effects that may bias GSMR results?

Minor points:

- Line 408, "merging all lead SNPs with $r^2 = 1$ to genomic loci", perhaps r^2 threshold here should be 0.1?
- It is interesting that in Table1 all genetic correlations involving Body fat% are 2-3 times stronger than the corresponding correlations of Fat-free mass. Is there some explanation to this?
- As an optional suggestion, it might be interesting to apply using GenomicSEM tool for joint (multivariate) analysis of the traits that show significant correlations.

Reviewers' comments:

Reviewer #1 (Remarks to the Author):

.....The study is interesting and incrementally adds to our understanding of body composition and psychiatric disorders. My major concerns involve the analytic methods (point 1 specifically).

1. Covariates. Covariates in the GWASs of body composition and related traits included the following covariates: 1) assessment centre, genotyping batch, smoking status, alcohol consumption, menopause, age, SES, 6 ancestral PCs. The rationalization for these covariates is unclear. In particular, it is unclear why variables such as smoking status and alcohol consumption are included. This would seem to be most clearly problematic for genomic correlation analyses subsequently evaluating GWASs of very clearly related phenotypes (e.g., e.g., smoking status covariate – genomic correlation with heavy smoking, alcohol consumption covariate – genomic correlation with alcohol dependence, SES and EDU). However, one could argue they may also be problematic for other covariates – e.g., might smoking be a mechanism linking psychiatric disorder to body composition? This appears inappropriate – e.g., it does not make sense to regress out the effects of alcohol consumption on body fat when examining genomic correlations between alcohol dependence and body fat - what is the genetic correlation between AD and body fat once one removes alcohol consumption? The authors are encouraged to rerun these analyses with PCs, age, genotyping batch, menopause, and assessment centre as covariates.

We thank the reviewer for raising this concern. We were primarily interested in the genetic variants that are associated with body composition independent of psychiatric traits, such as addiction. The rationale for the inclusion of smoking and alcohol consumption as covariates in our GWAS to derive body composition phenotypes that are free of confounding caused by addiction-related behaviours. However, we appreciate that this is a concern. Thus, as suggested, we present summary data for

tobacco and alcohol consumption in our sample in Supplementary Table 1 and we performed the GWAS analyses again excluding both addiction-related behaviours. We present the genetic correlations in Supplementary Table 11. The results are (very nearly) identical to the main analysis (p 10, para 1).

“Sensitivity analyses not adjusting the body composition GWAS for alcohol consumption or smoking yielded the same results (Supplementary Table 11).”

A similar rationale provided the basis for the inclusion of socioeconomic status (SES) as a covariate. We were interested in genetic variants that are associated with body composition independent of SES. To reduce potential confounding, we, therefore, covaried it. Additionally, the phenotypic correlation between SES and body fat percentage (BF%) was $r_p=0.02$ and between SES and fat-free mass $r_p=0.00$ in our subsample of the UK Biobank. It is therefore unlikely that exclusion of the covariate would have a major, or even a modest, effect on the results. Please, see the following plots for illustration. We, therefore, decided to retain the covariate.

Figure 1. Scatterplot of fat-free mass and socioeconomic status in the UK Biobank coloured by gender.

Figure 2. Scatterplot of fat-free mass and socioeconomic status in the UK Biobank coloured by gender.

To clarify this endeavour, we added the following to the introduction (p 6, para 1):

“We applied trait-specific illness- and medication filtering to obtain genomic variants associated with body composition traits independent of the confounding effects of somatic diseases, such as diabetes or endocrine illness, and addiction-related behaviors, including smoking and alcohol consumption, as well as psychiatric disorders.”

And to the methods section (p 19, para 1):

“We applied trait-specific medication and illness filtering to exclude participants with compromised hydration status and medications or illnesses known to affect body composition to identify genetic variation association with body composition phenotypes that are not confounded by illnesses and their downstream effects or metabolism-changing medication. We applied stringent exclusion criteria and covaried for addictive behaviour-related phenotypes, including smoking and alcohol consumption (for exclusion criteria, see Supplementary Table 3)”

2. Given that many other studies of the UK biobank use 20 PCs, justification for using 6 is needed.

Six principal components (PCs) were used because investigations suggested this was necessary to control for geographical variation in the dataset, and that increasing the number of PCs had negligible additional benefit. We additionally covaried for assessment centre to capture potential population stratification. To clarify our procedure we added the following (p 19, para 1).

“We performed GWAS including incremental numbers of principal components and checked each GWAS for inflation by calculating its LDSC intercept. We identified six principal components as the optimal number to adjust for population stratification within the European subsample and to not overcorrect the analysis retaining the greatest signal. Additionally, we included assessment centre as a covariate to adjust for population stratification.”

3. MR Pleiotropy Additional information should be provided on how the GSMR mendelian randomization analyses were conducted and in particular how these analyses addressed pleiotropy (For example was HEIDI-outlier used?).

We apologise for the sparse description of the method and hope that the following additions clarify this sufficiently (p 21-22). We also added the exact number of excluded pleiotropic SNPs due to the HEIDI outlier analysis and its p value in Supplementary Tables 8 and 9.

“Mendelian randomization is a method that uses genetic variants as instrumental variables, which are expected to be independent of confounding factors, to test for causative associations between an exposure and an outcome⁹⁴. Mendelian randomization can be used to infer credible causal associations when randomized-controlled trials are not feasible or are unethical⁹⁴. GSMR performs a multi-SNP Mendelian randomization analysis using summary statistics. Let z be a genetic variant (e.g., SNP), x be the exposure (e.g., psychiatric disorder) and y be the outcome (e.g., body composition trait). The basic idea of GSMR is that, if there are multiple independent (or nearly independent) SNPs (z) associated with x and the effect of x on y is causal, then all the x -associated SNPs will have an effect on y through x . In this case, b_{xy} at any of the x -associated SNPs is expected to be identical in the absence of pleiotropy as all the SNP effects on y are mediated by x . Therefore, increased statistical power can be achieved by integrating the estimates of b_{xy} from all the x -associated SNPs using a generalized least squares (GLS) approach^{38,96}. The GSMR method essentially implements summary data-based Mendelian randomization (SMR) analysis for each SNP instrument individually, and then integrates the b_{xy} estimates of all the SNP instruments by GLS, accounting for the sampling variance in both b_{zx} and b_{yz} for each SNP and the LD among SNPs. We used individual-level genotype data from a subsample of the anorexia nervosa GWAS to approximate the underlying LD structure to account for LD between the variants in the multi-SNP instrument. Pleiotropy is an important potential confounding factor that could bias the estimate and often results in an inflated test-statistic in Mendelian randomization analysis. We also removed potentially pleiotropic SNPs (i.e., SNPs that have effects on both risk factor and outcome) from this analysis using the heterogeneity in dependent

instruments (HEIDI) outlier method^{98,95} that detects pleiotropic SNPs at which the estimates of b_{xy} are significantly different from expected under a causal model. The power of detecting a pleiotropic SNP depends on the sample sizes of the GWAS data sets and the deviation of b_{xy} estimated at the pleiotropic SNP from the causal model. Based on this, the overall b_{xy} can be estimated from all the instruments remaining using a generalized least squares approach that takes the LD between the variants and the correlations between their effect sizes into account by modeling them in a covariance matrix. Additionally, GSMR uses the intercept of the bivariate LD score regression to account for potential sample overlap between the GWAS used as instruments for the exposure or outcome¹². Estimates with binary exposures were converted to the liability scale⁹⁷.”

4. Minor comment: The adjusted Bonferroni multiple correction testing for body composition/physical activity and glycemic traits analyses independently appear appropriate given the correlated nature of many interested variables. However, addition description/clarification i.e., how many independent traits went into the matrix decomposition, would be useful.

We added the following description to clarify the generation of the matrices (p 23, para 2):

“The first matrix of the main analysis contained all 17 psychiatric traits, all four body composition traits, physical activity, and childhood obesity (Supplementary Table 6). All sex-specific correlations were entered when available. The second matrix comprised all 17 psychiatric traits and all glycemic traits listed in Supplementary Table 10, including their sex-specific correlations.”

Reviewer #2 (Remarks to the Author):

.....The work has been performed quite comprehensively and these findings generate new hypotheses for targeted preventive strategies. Below are my main comments followed by some minor comments.

1. The authors only mentioned sex-specific common variant genetic correlations of body composition, physical activity and glycemic traits with 17 psychiatric and behavioral traits in the abstract section, but did not mention the age-dependent genetic influences they observed between psychiatric traits and body composition. I would suggest that they should add a brief summary description in this section.

We thank the reviewer for this comment and have added the following sentence to the abstract:

“Education years and ADHD show genetic overlap with childhood obesity.”

2. The study is generally a statistics study which pinpoint the shared genetics between psychiatry disorders/behavioral traits and energy homeostasis/body composition traits. And the authors do find some associations between these traits. However, the authors should provide more biological interpretation in the discussion part. For example, ADHD have high genetic correlation with body fat %, the question is are there some shared biological pathways that responsible for this phenomenon?

As the reviewer correctly notes our analysis is primarily statistical. Therefore, it does not allow for the interpretation or identification of “shared biological pathways”. To circumvent this limitation, we included body composition-related traits in our analysis, including glycemic measures (e.g., fasting insulin and glucose concentrations) and physical activity that potentially may moderate or mediate the relationship between body composition and psychiatric traits. However, we only detected statistically significant genetic correlations between glycemic traits and anorexia nervosa. We added the following paragraph (p 13, para 2):

“Our finding of a genetic overlap between ADHD and obesity in childhood may implicate shared biological pathways between both traits. Given our other results, it appears that this shared component is unlikely to be related to physical activity or glycemic traits. Instead, speculatively, a central nervous system pathway that is dysregulated by increased body mass in childhood may increase the liability to develop ADHD.”

3. The introduction: On line 79 and 80, the authors stated that "...These analyses have shown substantial overlap between psychiatric disorders, providing evidence for an underlying "p factor" representing general liability for psychiatric illness ...", the authors introduced several different approaches to use genome wide summary statistics generated by GWAS but it is not clear what are substantial overlaps between psychiatric disorders. It should be clarified simply or provide some examples to demonstrate it.

We hope the following additions clarify our introductory paragraph (p 4, para 1):

"Such GWAS based genetic correlation analyses have shown substantial genetic overlap among psychiatric disorders¹³, providing evidence for an underlying "p factor" representing general liability for psychiatric illness^{14,15}. For instance, genomic structural equation modelling¹³ of GWAS summary statistics for schizophrenia, bipolar disorder, major depressive disorder, post-traumatic stress disorder, and anxiety showed that they load onto one shared latent factor with loading estimates between 0.29-0.86¹⁶."

4. Page 422: Section of "GWAS of psychiatric disorders and behavioral traits", it should be clarified in the online methods section, which psychiatric disorders and behavioral traits GWASs were used for sex- and age-dependent genetic correlations analysis.

To clarify, we added the following sentence (p 20, para 3):

"All of the following traits were used for the sex-specific and age-dependent analyses (Supplementary Table 3)."

5. More details on the generalized summary data-based Mendelian randomization (GSMR) methods are required in the online methods (on page 446). Specifically, it would be good to add a description of the principle of GSMR and the meaning of the parameters.

We thank the reviewer for this suggestion and extended the description. Please, see our response to Reviewer #1's point 3.

6. Now it reads as slightly too comprehensive and does not give a reality check. You just were able to show that five psychiatric disorders (anorexia nervosa, OCD, schizophrenia, ADHD, and alcohol dependence) as well as three behavioral traits (education years, insomnia, and heavy smoking) have significant genetic correlations (i.e., shared genetics) with body composition in two distinct patterns, however, the main drawback with the current version of the manuscript is that it is not easy to understand what you actually found or what are the specific applications of these genetic correlations. And the Table 1 tries to summarize this, but is so comprehensive that it does not highlight the main findings. This should be clarified.

The main Table 1 in the manuscript that shows 16 genetic correlations is a short summary of Supplementary Table 6 that includes all 703 genetic correlations. We have clarified this in the table caption. Additionally, we are giving an overview over our findings in Figure 1.

"The table presents the significant findings. The full results can be found in Supplementary Table 6."

Furthermore, our study shows the following key findings apart from the genetic correlations between body fat % and fat-free mass with psychiatric traits:

- Anorexia nervosa shows a stronger genetic correlation with body fat % in females than in males. This may partially explain the sex-dependent liability to anorexia nervosa (p 12, para 1).

- The genomic overlap between body composition traits and psychiatric disorders occurs in later adulthood. Childhood and young adulthood GWAS of body mass index do not correlate with psychiatric traits (p 12, para 2).
- Objectively-measured physical activity assessed by accelerometers shows genetic correlations only with obsessive-compulsive disorder and anorexia nervosa but with no other psychiatric disorder. This is especially unexpected as we hypothesised that the hyperactivity seen in attention-deficit/hyperactivity disorder (ADHD) may have a genetic underpinning (p 13, para 2).
- Insulin traits that may mediate the relationship between psychiatric and body composition traits showed significant genetic correlations only with anorexia nervosa and years of education which positions anorexia nervosa as a unique disorder amongst the investigated psychiatric disorders (p 14, para 2).

Minor comments

1. The current version of this manuscript needs to be checked for logic and you need to pay attention to the typesetting of the text. For example on page 411 the authors forgot the blank line.

We thank the reviewer for spotting this. We have rechecked the manuscript to ensure there are no further omissions. We have added the missing blank line.

2. We try to get access to the URL <https://github.com/topherhuebel/>, but the code used for analysis in this research is not uploaded yet. It is highly recommended to upload the source code for further review.

We apologise for not providing the path to the correct subfolder (ukbgwas). To reassure the reviewer, the scripts and the analysis plan have been completely available since 2 November 2018 and the

folder had the following description: "Scripts for body composition GWAS in UK Biobank". We have extended the link in the manuscript (p 24, para 1).

<https://github.com/topherhuebel/ukbgwas>"

Reviewer #3 (Remarks to the Author):

This article conducts a cross-trait analysis of summary statistics for psychiatric disorders, body composition, glycemic traits and physical activity, using LDSR for genetic correlation, GSMR for mendelian randomization, and several other well-established methods for the supplementary analyses. The manuscript is very well written, and the results provide novel scientific insights. Particularly, it is intriguing that the genetic correlation between schizophrenia and body fat% is negative, against what one would expect from the epidemiological studies. Similarly, it is interesting that there is a genetic relationship between ADHD/EDU and overweight before the age of ten, and a lack of genetic correlation between ADHD and physical activity. These findings are well interpreted in the discussion section and might have important clinical implications.

We thank the reviewer for the feedback.

I only have a few comments:

1. Please revise Fig2 and the corresponding description of GSMR results, as it has several inconsistencies:

We thank for these comments and have updated Figure 2 and its description.

- Line 171: "1.9 kg decrease in fat mass per year of education", is this statement correct? Here 1.9 seems like a very large effect, at least if EDU is measured in the actual years (which I would expect to range somewhere from 10 to about 20)?

The GWAS of education years was indeed measured in years. We were also surprised by the effect size of the findings for education years. Updating our GSMR analysis with the latest GCTA software release, including the updated HEIDI outlier analysis, resulted in a comparable (although increased) estimate of 3.74 kg decrease in fat mass per year of education. This analysis was one of our best powered GSMR analyses as we were able to include 47 SNPs as instrumental variables, even after 20 SNPs were removed due to potential pleiotropy.

- Fig2C, EDU years are shown as "odds ratio" which is inconsistent with Supplementary Table 3 where EDU is shown as a continuous trait.

We thank the reviewer for this comment and apologise for this error. We have re-run all GSMR analysis with the most recent version of the GCTA software and used the most up to date HEIDI outlier analysis to test for pleiotropy in the SNPs that were included in the multi-SNP instrument. Accordingly, we revised the figure and its description.

- Lines 163-165: The description says "Odds Ratio (Fig 2A, 2B), Beta (Fig 2C)" which is inconsistent with the actual Fig2, which shows Beta on Fig 2A, 2B, and Odds Ratio on Fig 2C

We apologise for the mislabeling and thank the reviewer for noticing. We have corrected the figure caption and the in-text information.

2. It would be helpful to expand the discussion around sex-specific genetic correlations, i.e. Anorexia nervosa (AN) and Years of education (EDU) versus body fat%. Is there some way to interpret these findings? If I understood correctly, the results stand for correlation of sex-specific body fat% versus non-sex-specific AN and EDU. Would it be possible to conduct a similar analysis of sex-specific EDU and sex-specific AN versus (also sex-specific) body fat% ?

We thank the reviewer for this suggestion and have added the following to the discussion (p 12, para 2).

“In our analysis, anorexia nervosa showed a stronger correlation with body fat % in females than in males. This phenomenon was not observed for other traits genetically associated with anorexia nervosa, such as neuroticism, anxiety, major depressive disorder, OCD, or schizophrenia⁴¹. These findings suggest that anorexia nervosa and body fat % may share a sex-dependent set of genomic variants potentially contributing to its marked sex bias in its prevalence. Education years showed a stronger genetic correlation with fat mass in males than in females. However, the GSMR analysis showed a more pronounced protective effect of education years on fat mass in females than in males in line with a large epidemiological study⁴⁴. This suggests that the stronger genetic association between education years and fat mass in males may be driven by a set of pleiotropic variants.”

For the genetic correlations with female-only anorexia nervosa and female- and male-only education years, we refer to Supplementary Table 6. Using sex-specific GWAS of psychiatric traits resulted in similar results to those observed with the sex-combined GWAS of psychiatric traits.

3. I suggest adding few words about the limitations of the GSMR. For example, is there sample overlap between GWAS studies in the analysis that may affect GSMR results? Is there an indication of the pleiotropic effects that may bias GSMR results?

We thank the reviewer for these comments. We have extended the paragraph on the GSMR method (p 21-22). The method takes sample overlap into account and checks for potential pleiotropic variants and excludes them from the analysis. Please, see our response to Reviewer #1's point 3.

Minor points:

- **Line 408, "merging all lead SNPs with $r^2 = 1$ to genomic loci", perhaps r^2 threshold here should be 0.1?**

We thank you for picking this up. We have corrected it to $r^2 = 0.1$.

- **It is interesting that in Table1 all genetic correlations involving Body fat% are 2-3 times stronger than the corresponding correlations of Fat-free mass. Is there some explanation to this?**

As we previously reported (Hübel et al. 2018), the heritability of fat-free mass in our subsample of the UK Biobank was enriched for connective, bone, adrenal, and pancreas tissue whereas the heritability of body fat % in females was enriched for the central nervous system. The overlap between genetic variants that are enriched for central nervous tissue with psychiatric traits may be a potential explanation for the stronger genetic correlations.

- **As an optional suggestion, it might be interesting to apply using GenomicSEM tool for joint (multivariate) analysis of the traits that show significant correlations.**

We thank the reviewer for this suggestion. We are currently working on follow up analyses including GenomicSEM. This will form part of a subsequent paper.

Reviewers' Comments:

Reviewer #2:

Remarks to the Author:

The authors have addressed all of my questions.

Reviewer #3:

Remarks to the Author:

The authors addressed my points, including corrected Figure 2, and a detailed description of GSMR method. I don't have further comments.

Update from Aug 9th, 2019. Additionally I was asked by the Editor to comment on authors response to concerns raised by Reviewer #1. In my opinion authors fully addressed these concerns in their revision.

RE covariates, authors explained the rationale for including smoking, alcohol consumption and socioeconomic status as covariates, and shows in Suppl.table 11 that it has only a minor effect of genetic correlations. I agree with retaining these covariates in the main analysis.

RE other questions, authors provided justification for their choice of 6 PCs as covariates, substantially expanded their description of GSMR method, and gave sufficient details about their procedure to estimate the number of effective tests in multiple correction.

Reviewers' comments:

Reviewer #2 (Remarks to the Author):

The authors have addressed all of my questions.

We thank the reviewer for taking the time to review our manuscript and the constructive suggestions.

Reviewer #3 (Remarks to the Author):

The authors addressed my points, including corrected Figure 2, and a detailed description of GSMR method. I don't have further comments.

Update from Aug 9th, 2019. Additionally I was asked by the Editor to comment on authors response to concerns raised by Reviewer #1. In my opinion authors fully addressed these concerns in their revision.

RE covariates, authors explained the rationale for including smoking, alcohol consumption and socioeconomic status as covariates, and shows in Suppl.table 11 that it has only a minor effect of genetic correlations. I agree with retaining these covariates in the main analysis.

RE other questions, authors provided justification for their choice of 6 PCs as covariates, substantially expanded their description of GSMR method, and gave sufficient details about their procedure to estimate the number of effective tests in multiple correction.

We are grateful that the reviewer was able to take the time to review our manuscript and the constructive suggestions provided and the additional time to respond to the concerns raised by Reviewer #1.